# Functionalized Au$_{15}$ nanoclusters as luminescent probes for protein carbonylation detection

Guillaume F. Combes [1,2], Hussein Fakhouri[1,3], Christophe Moulin [3], Marion Girod[4], Franck Bertorelle[3], Srestha Basu[3], Romain Ladouce[2], Martina Perić Bakulić [1], Željka Sanader Maršić [5], Isabelle Russier-Antoine[3], Pierre-François Brevet [3], Philippe Dugourd[3], Anita Krisko[6], Katarina Trajković[1,2], Miroslav Radman [1,2,7], Vlasta Bonačić-Koutecký [1,8,9 ✉] & Rodolphe Antoine [3 ✉]

Atomically precise, ligand-protected gold nanoclusters (AuNCs) attract considerable attention as contrast agents in the biosensing field. However, the control of their optical properties and functionalization of surface ligands remain challenging. Here we report a strategy to tailor AuNCs for the precise detection of protein carbonylation—a causal biomarker of ageing. We produce Au$_{15}$SG$_{13}$ (SG for glutathione) with atomic precision and functionalize it with a thiolated aminooxy moiety to impart protein carbonyl-binding properties. Mass spectrometry and molecular modelling reveal the key structural features of Au$_{15}$SG$_{12}$-Aminooxy and its reactivity towards carbonyls. Finally, we demonstrate that Au$_{15}$SG$_{12}$-Aminooxy detects protein carbonylation in gel-based 1D electrophoresis by one- and two-photon excited fluorescence. Importantly, to our knowledge, this is the first application of an AuNC that detects a post-translational modification as a nonlinear optical probe. The significance of post-translational modifications in life sciences may open avenues for the use of Au$_{15}$SG$_{13}$ and other nanoclusters as contrast agents with tailored surface functionalization and optical properties.

[1] Center of Excellence for Science and Technology-Integration of Mediterranean Region (STIM), Faculty of Science, University of Split, Split, Croatia. [2] Mediterranean Institute for Life Sciences (MedILS), Split, Croatia. [3] Univ Lyon, Univ Claude Bernard Lyon 1, CNRS, Institut Lumière Matière, Villeurbanne F-69622, France. [4] Univ Lyon, CNRS, Université Claude Bernard Lyon 1, Institut des Sciences Analytiques, UMR 5280, 5 rue de la Doua, Villeurbanne F-69100, France. [5] Faculty of Science, University of Split, Split, Republic of Croatia. [6] Department of Experimental Neurodegeneration, University Medical Center Goettingen, Göttingen, Germany. [7] Université R. Descartes-Paris 5, Faculté de Médecine, site Cochin, Paris, France. [8] Interdisciplinary Center for Advanced Science and Technology (ICAST) at University of Split, Split, Croatia. [9] Chemistry Department, Humboldt University of Berlin, Berlin, Germany. ✉email: vbk@cms.hu-berlin.de; rodolphe.antoine@univ-lyon1.fr

Proteins carry out and assure the maintenance of almost all cellular functions, lending support to a paradigm arguing that aging and age-related diseases (ARDs) are complex consequences of the cumulative oxidative damage to proteins[1,2]. Previous research has gathered evidence supporting the hypotheses that healthy aging is an increasing biological noise, a consequence of diffuse proteome oxidation, whereas ARDs appear associated with excessive oxidation of particular susceptible proteins sensitized by mutations predisposing to disease[2]. Protein carbonylation, an irreversible oxidative damage to proteins can affect most amino acids[3]. Indeed, an increase in protein carbonyls appears as a biomarker of cellular and organismal aging[4].

Different methods have been developed for the detection and quantification of carbonylated proteins as commonly used markers of protein oxidation[4–7]. Since protein carbonyls have no distinguished UV or visible optical properties, specific chemical probes are required for their visualization[6,7]. The use of fluorophores with carbonyl-reactive groups enables direct detection and quantitation of carbonyls on the proteins subjected to one- and two-dimensional electrophoresis (1DE and 2DE, respectively) using fluorescence imaging scanner[4]. For instance, cyanine hydrazide is currently used in the two-dimensional gel electrophoresis methodology. For accurate detection, this complex analysis requires special equipment and reagents[8–10]. Moreover, the hydrazone bond formed between the cyanine hydrazide and protein carbonyls is usually sensitive to acidic environment and narrow pH range conditions are required to optimize the efficient binding of cyanine-hydrazide fluorescent dye to its carbonyl targets. Recently, it has been reported that near-infrared (NIR) fluorescence using NIR dyes offers certain advantages over visible-range fluorescence, particularly the lack of autofluorescence of biological molecules in the NIR[11]. To improve the specificity and sensitivity of carbonyl detection necessary for diagnostics and prognostic purposes, there is room for advanced strategies.

One promising strategy for the detection of biomolecules consists of using nonlinear optical processes (NLO) involving multiple IR photon excitations (in the NIR-IR window, i.e. 700–800 nm)[12,13]. Also, the high spatial resolution of two-photon absorption (2PA or TPA) is a strong added value for bioimaging applications[14]. Ligand-protected gold nanoclusters (AuNCs) with gold kernel composed of few dozen of atoms possess molecule-like properties[15] such as luminescence[16,17]. Such AuNCs can exhibit strong photoluminescence from ultraviolet to the NIR region[18,19]. Also, their bleaching rate is very slow, demonstrating their superior photostability[20,21]. In addition, ligand-protected AuNCs exhibit outstanding biocompatibility, which makes their in vitro and in vivo bio-applications a rich research area[22–27]. As compared to dyes, ligand-protected AuNCs present much greater two-photon absorption cross sections[13,28,29], making them promising candidates for multiphoton excited fluorescence microscopy[30–32]. As pioneered by Murray and co-workers[33], surface functionalization of AuNCs, in particular their functionalization through ligand-exchange strategy[34], grants them high versatility, while selective functionality is incorporated onto the ligand-protected NCs by exchanging the surface protecting ligand with desired molecules containing appropriate functional groups.

In this work, we conducted the proof-of-concept study for developing the first NC-based imaging system for protein carbonylation detection. The NCs were liganded with glutathione (SG) and produced at the atomic precision with the exact formula $Au_{15}SG_{13}$. Such NCs were then functionalized with a thiolated aminooxy probe to gain protein carbonyl-binding properties. Using mass spectrometry (MS) approach, we then showed that the resulting NCs bind carbonylated proteins through the formation of an oxime bond between the aminooxy-containing thiolated ligand on the NC and the carbonylated amino acid on

the protein. Molecular modeling was performed to reveal the key features of functionalized NCs and to evaluate the robustness of the oxime bond upon exposure to the solvent. Finally, we demonstrate that such functionalized AuNCs can detect protein carbonyls in gel-based 1DE analysis by one-photon fluorescence and two-photon excited fluorescence imaging.

## Results

**Synthesis and characterization of $Au_{15}SG_{13}$ and $Au_{15}SG_{12}$-Ao NCs.** Here, we have developed a synthetic protocol to produce atomically precise thiolated aminooxy-functionalized gold NCs with protein carbonyl-binding properties. Initially, non-functionalized, glutathione-protected gold NCs ($Au_{15}SG_{13}$) were synthesized (Fig. 1a). We have chosen $Au_{15}SG_{13}$ as a basis for further modifications due to its small size, simplicity of its synthesis, good stability in water, and excellent optical properties in dried polyacrylamide matrix. FT-IR spectra of $Au_{15}SG_{13}$ NCs and pure glutathione GSH are given in Supplementary Fig. S1a, b. The ligation of glutathione in the form of the thiolate (SG) to the Au core was confirmed by the absence of the absorption band at $\nu(S-H) = 2523$ cm$^{-1}$ in the FTIR spectrum of the as-prepared NCs sample, as already found in the seminal work published by Negishi and Tsukuda[35]. Supplementary Fig. S1c shows the TEM image of the as-prepared $Au_{15}SG_{13}$ NCs. The particles with sizes of 1–2 nm are barely discernible in the image. From the XPS data (Supplementary Table S1), we find the Au/S atomic ratio to be $1.26 \pm 0.13$, which is compatible with the composition of $Au_{15}SG_{13}$ (the expected value is 1.15). Upon synthesis, the quality of $Au_{15}SG_{13}$ was assessed using negative-mode ESI-MS. This analysis confirmed monodispersity of the synthesized NCs. A charge state distribution was observed from $[M-4H]^{4-}$ through $[M-6H]^{6-}$. Deconvolution of charge states from 4- through 6- revealed a mass of 6928 Da for the $Au_{15}SG_{13}$, consistent with its calculated mass (Supplementary Fig. S2a).

To develop NCs with carbonyl-binding properties, $Au_{15}SG_{13}$ had to be functionalized by the replacement of one glutathione with a carbonyl-reactive agent. Hydrazides (Hz) and aminooxy (Ao) are commonly used carbonyl-reactive chemical groups and they form different kinds of bonds with protein carbonyls–hydrazone (Hz) and oxime (Ao) bonds. Since oxime bonds appear to be more stable than hydrazone bonds[36,37], we opted for the Ao to functionalize $Au_{15}SG_{13}$. Functionalized NCs $Au_{15}SG_{12}$(3-Aminooxy)-1-propanethiol (termed as $Au_{15}SG_{12}$-Ao) was generated by the replacement of one glutathione on $Au_{15}SG_{13}$ with the Ao through a ligand-exchange procedure (Fig. 1b).

To characterize the synthesized $Au_{15}SG_{12}$-Ao, the reaction mix was analyzed using negative-mode ESI-MS and a new peak corresponding to $Au_{15}SG_{12}$-Ao for the charge state 4- was observed, as expected (Fig. 1c). Experimentally determined isotopic patterns of the different NCs were in perfect agreement with their simulated ESI-MS patterns (Supplementary Fig. S2b, c) and confirmed the stoichiometry of $Au_{15}SG_{13}$ (Supplementary Fig. S2b) and $Au_{15}SG_{12}$-Ao (Supplementary Fig. S2c). Of note, increasing the concentration of the Ao led to the exchange of more than one SG ligand (Fig. 1d and Supplementary Fig. S2d, e). ESI-MS was also applied to monitor the number of Ao ligand exchanged in $Au_{15}SG_{13}$ species following the addition of Ao in solution at different concentrations. Supplementary Fig. S2d, e shows the evolution in ligand exchange as a function of the concentration of Ao ligand. Clearly, adding 0.1–0.3 equivalent of Ao allows for controlling of ligand exchange to just one.

We next compared UV-vis absorption and emission spectra of $Au_{15}SG_{13}$ and $Au_{15}SG_{12}$-Ao (Supplementary Fig. S2f). The two NCs displayed similar main features of the spectra. The linear optical absorption spectra were composed of a monotonous

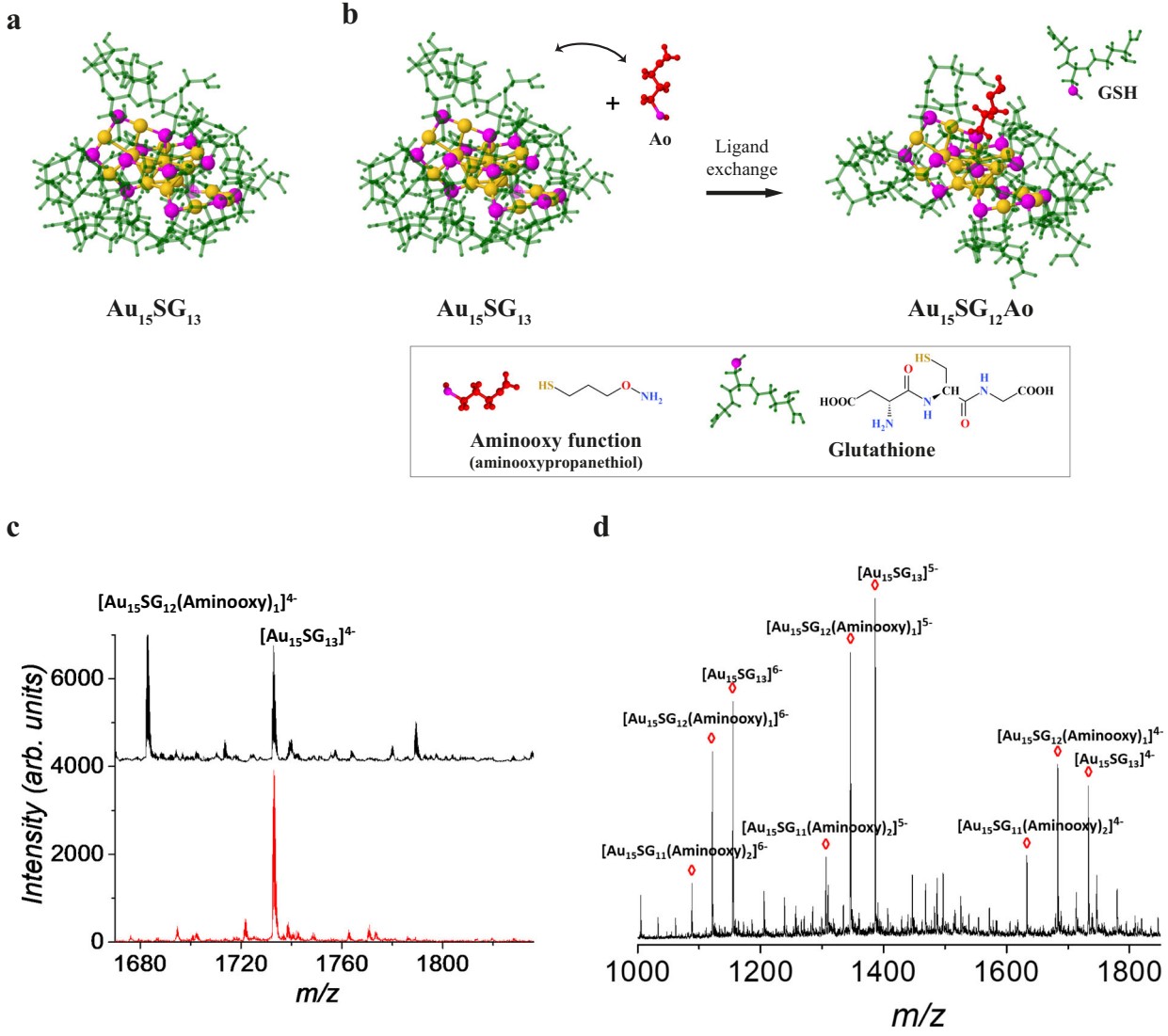

**Fig. 1 Synthesis and characterization of Au₁₅SG₁₃ and Au₁₅SG₁₂-Ao. a** Density functional theory (DFT) structure of the gold NCs (Au₁₅SG₁₃). **b** DFT structures illustrate the ligand-exchange strategy to functionalize the NCs with an aminooxy (Ao) function (Au₁₅SG₁₂-Ao). Atoms and molecules are labeled as following: S-magenta, Au-gold, glutathione-green, and aminooxypropanethiol-red **c** Zoom of the ESI mass spectrum of Au₁₅SG₁₃ and Au₁₅SG₁₂Ao NCs corresponding to the *m/z* region labeled by a blue rectangle in supplementary Fig. S2a. A new peak corresponding to Au₁₅SG₁₂-Ao is observed following the ligand-exchange reaction. **d** Mass spectra showing exchange of more than one SG ligand.

increase of absorption below 500–550 nm. Photoluminescence spectra displayed a broad band extending in the NIR region and centered around 650−700 nm.

To determine the structure of NC after the ligand exchange, we performed molecular modeling based on the combination of density functional theory (DFT) and semi-empirical quantum method PM7 approach (see Computational details). The structure of Au₁₅SG₁₃ was proposed by De-en Jiang[38] using the density functional theory (DFT). According to this model, Au₁₅SG₁₃ contains a cyclic [Au(I)-SG] pentamer interlocked with two trimer motifs protecting the tetrahedral Au₄ core. Such structural assignment was supported by comparison to the powder X-ray diffraction pattern and, via time-dependent DFT calculations, to the optical and chiroptical (CD) absorption spectra[39]. We evaluated the influence of the Ao position on the overall structure of the liganded Au₁₅SG₁₃ NC. Two possible exchanges out of thirteen are presented in structures I and II (Supplementary Fig. S2g). In both structures, H-bond networks between neighboring SG ligands and the Ao ligand are present. Interestingly, the Ao in structure I is more buried in the SG ligand environment than the

Ao in structure II. Such differences in Ao accessibility may influence their reactivity with carbonyls.

**Interaction of Au₁₅SG₁₂-Ao with protein carbonyls.** Since protein carbonylation occurs on solvent-exposed amino acids, we predicted that Au₁₅SG₁₂-Ao would react with carbonylated proteins through the formation of an extremely stable oxime bond between the amino group of the Ao attached to the NC and carbonyl groups on the amino acids (Fig. 2a). To test this prediction empirically, we used two model substrates: leupeptin (N-acetyl-L-leucyl-L-leucyl-L-argininal), a natural tripeptide inhibitor of serine proteases containing a carbonyl group, and oxidized lysozyme as a model protein.

**Interaction of Au₁₅SG₁₂-Ao with leupeptin.** If Au₁₅SG₁₂-Ao can bind protein carbonyls, we would expect the formation of a stable [Au₁₅SG₁₂-Ao–leupeptin] complex upon mixing of the Au₁₅SG₁₂-Ao with leupeptin (Fig. 2b). To get insight into the formation of such complex and test specificity of Au₁₅SG₁₂-Ao for a carbonyl on

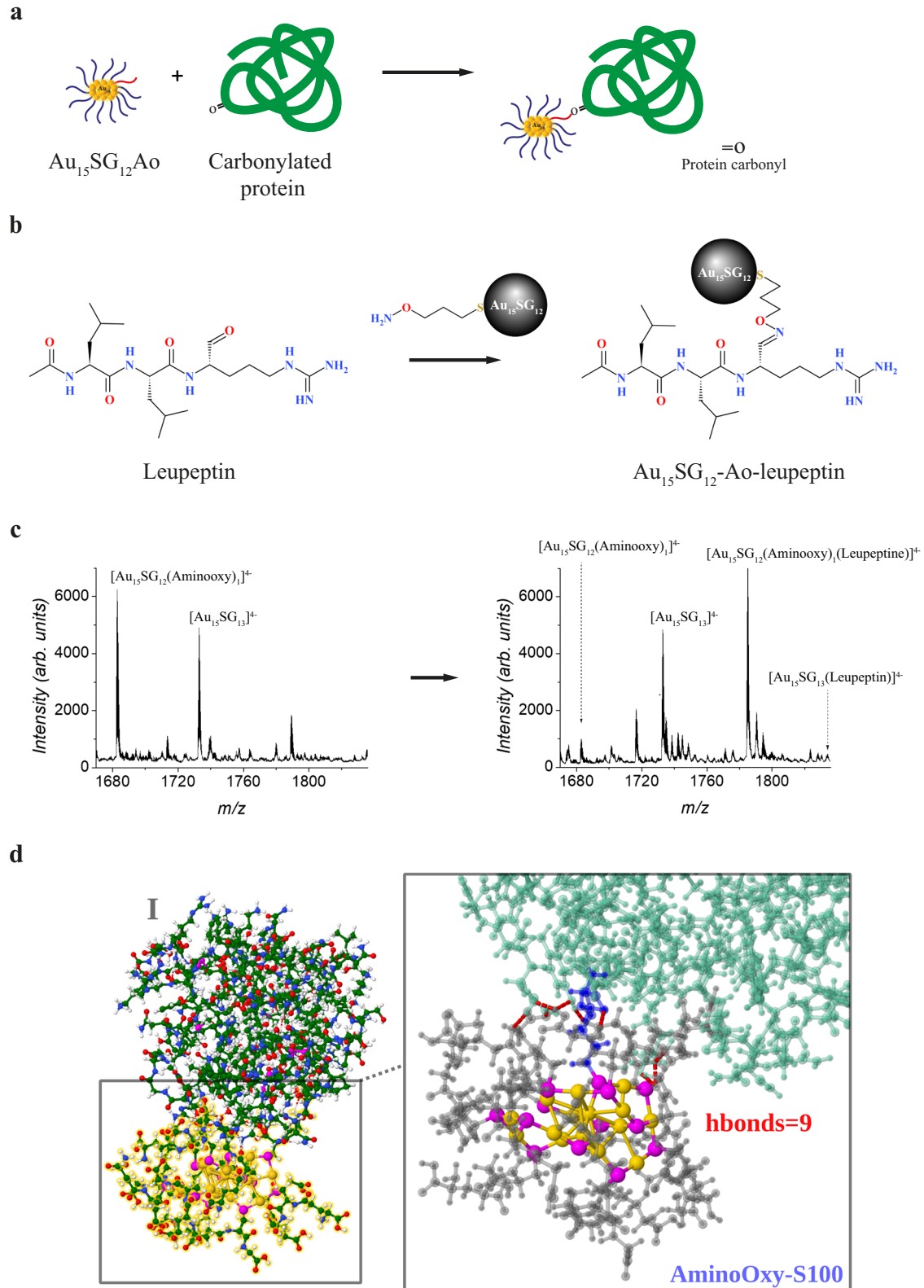

**Fig. 2 Au$_{15}$SG$_{12}$-Ao binds to protein carbonyls. a** Schematic illustration of the functionalized gold NCs (Au$_{15}$SG$_{12}$-Ao) binding to a carbonylated protein. **b** Illustration of the reaction of the Ao group of 3-(Aminooxy)-1-propanethiol forming an oxime linkage with the aldehyde of leupeptin. **c** ESI-MS spectrum following the formation of the stable oxime linkages: a new peak corresponding to Au$_{15}$SG$_{12}$-Ao-leupeptin is observed. **d** The QM/MM structures obtained by using two-layer ONIOM illustrating the interaction of liganded AuNC with lysozyme. Interface between liganded NCs Au$_{15}$SG$_{12}$-Ao and the protein is enlarged (**right**) illustrating Ao-serine oxime bond and hydrogen bonding network. (SG-gray, Au-gold, S-magenta, lysozyme-green, Ao-Serine oxime bond-blue, H-bonds-dotted in red).

leupeptin, a mixture of $Au_{15}SG_{13}$ and $Au_{15}SG_{12}$-Ao was analyzed by negative-mode ESI-MS before and after addition of leupeptin (Fig. 2c). As predicted, after the addition of leupeptin a new peak was detected under the charge state 4-, which corresponded to the newly formed [$Au_{15}SG_{12}$-Ao-leupeptin] complex. Note that the precursor $Au_{15}SG_{12}$-Ao has almost disappeared demonstrating its high reaction rate with leupeptin. Importantly, the $Au_{15}SG_{13}$ peak was unchanged, indicating that the non-functionalized NC does not react unspecifically with leupeptin under indicated conditions.

**Characterization of oxidized lysozyme.** We next aimed to test whether $Au_{15}SG_{12}$-Ao reacts specifically with amino acids carbonylated upon oxidation of a protein. To that end, we performed a multifaceted analysis of the NC-protein complex using oxidized lysozyme as a model protein. Lysozyme is a single chain polypeptide of 129 amino acids that we used previously in systematic protein carbonylation studies[40]. To induce carbonylation of the lysozyme we chose the metal-catalyzed oxidation (MCO). The MCO occurs in vitro by Fenton reaction whereby ascorbic acid and iron-chloride generate highly reactive oxygen species (ROS such as hydroxyls) that react with proteins and generate carbonyl groups on susceptible amino acids[5]. To confirm the efficiency of the MCO, we measured carbonylation of the lysozyme by a Western Blot-based method (Supplementary Fig. S3a, Supplementary Data 1 and 2 Figs. S8, S14) and by a quantitative 2,4-Dinitrophenylhydrazine (DNPH) colorimetric assay (Supplementary Fig. S3b and Supplementary Data 2 Fig. S15)[41]. Both approaches revealed a significant increase in carbonylation in oxidized lysozyme as compared to non-oxidized control under indicated conditions.

To identify and quantify carbonylated amino acids on the oxidized lysozyme, sequence database search and tandem mass spectrometry MS/MS analysis were performed (Supplementary Fig. S3c). To search for oxidized peptides, we implemented user-defined variable modifications corresponding to a list of known carbonyl modifications from the literature[42], which were derivatized with DNPH (Oxi-DNP method). For the MS/MS analysis, we labeled carbonylated amino acids with DNPH since DNPH-tagged-carbonylated peptides display the best results with this method. Namely, DNPH tag has better ionization efficiencies and additionally stabilizes labile modifications such as Michael adducts as compared to alternative aminooxy probes[43]. The combined digest data provided 61.2% sequence coverage for the lysozyme subjected to MCO. Nineteen Oxi-DNP modified peptides were identified with confidence in the derivatized MCO-lysozyme (Supplementary Fig. S3c). Among the identified carbonylation sites, tryptophans (W) appeared as the most frequently oxidized amino acids. As a control, same analysis was performed on a non-oxidized lysozyme treated with DNPH and no modified peptides were detected.

Quantification of detected oxidized peptides has been done at the $MS^1$ level (Supplementary Fig. S3c). To estimate the amount of oxidized versus the non-oxidized protein peak areas for each oxidized peptide were normalized to the areas of the corresponding non-oxidized peptides in the control samples (Supplementary Fig. S3d). Approximately 1.3 % of the total protein amount has been oxidized, with $W^{108}$, $K^{13}$, and $W^{62}$ as the most intense carbonylated sites.

**Molecular modeling of lysozyme–$Au_{15}SG_{12}$-Ao complex revealing the key role of liganded AuNC.** To visualize the positions on the protein of the empirically determined carbonylated sites (Supplementary Fig. S3c), those sites ($K^{13}$, $S^{24}$, $W^{28}$, $T^{43}$, $W^{62}$, $P^{79}$, $S^{81}$, $L^{83}$, $S^{85}$, $I^{98}$, $S^{100}$, $W^{108}$, $Q^{121}$, $W^{123}$, and $I^{124}$) were mapped on the 3D structure of lysozyme obtained by X-ray analysis (Supplementary Fig. S3e)[44]. While most of these residues were found on the protein surface and are thus easily accessible to

ROS, the carbonylated tryptophan residues $W^{28}$ and $W^{123}$ were buried inside the protein skeleton. This is likely due to protein misfolding caused by the initial carbonylation of the surface residues and subsequent exposure to ROS of the previously hidden parts of the polypeptide.

To characterize the linkage between $Au_{15}SG_{12}$-Ao and carbonylated amino acid, we next conducted molecular modeling on an example of $S^{100}$ on the surface of the lysozyme (Fig. 2d). Of note, the modeling has been performed on non-oxidized protein because the structure of carbonylated lysozyme is not available. In order to include the natural environment of the $Au_{15}SG_{12}$-Ao–oxidized lysozyme complex, we also evaluated the robustness of the oxime bond towards solvent accessibility. This analysis revealed that glutathione ligands play the protective role with respect to the thiolated aminooxy ligand while allowing it to form the interface with carbonylated lysozyme. Penetration of water was significantly low suggesting that the Ao linkage is protected from the external environment by the glutathione surrounding (Supplementary Fig. S3f). The key result is the H-bond network formed by glutathione ligands that protect the oxime bond between the Ao and a carbonylated residue on the protein. Together, these data justify the use of liganded AuNCs for the detection of protein carbonylation.

**MS-based detection of $Au_{15}SG_{12}$-Ao grafted on carbonylated amino acids within the oxidized lysozyme.** To obtain evidence that $Au_{15}SG_{12}$-Ao is specific for carbonylated amino acids, i.e. that it is grafted directly on the carbonylated amino acid residue on the oxidized lysozyme, we next made attempts to measure directly the mass of $Au_{15}SG_{12}$-Ao bound to oxidized lysozyme. However, due to low levels of carbonylation (<1%) no NC attached to the oxidized lysozyme was detected by mass spectrometry (MALDI-MS technique), as evidenced by lack of any mass peak larger than that of the parent oxidized lysozyme. Hence we used an alternative approach where we analyzed the oxidized protein derivatized with $Au_{15}SG_{12}$-Ao after subsequent degradation of the grafted NC and trypsin digestion, whereby the putative oxime bond between the aminooxy on the NC and carbonylated amino acids on the protein remained intact. Degradation of the NC was necessary since NC-grafted peptides are too large to be analyzed directly by LC-MS/MS and it was achieved by cysteine treatment which destabilizes bonds between the glutathiones and the gold core of the NC. The modified peptides were obtained after neutralization of ungrafted Ao groups and precipitation of the degraded gold-cysteine polymers. After this treatment, the residual modification on the grafted carbonylated sites should be an Ao-C3-thiol group. The sample was then reduced by dithiothreitol and alkylated with iodoacetamide (IAM) before digestion with trypsin. As the IAM can react with the free thiols remaining on the carbonylated sites, the carbonyl modifications[5] derivatized with Ao-C3-thiol-IAM were implemented for the database search. Seven peptides with these modifications were identified using Protein prospector (Table 1). We found modifications corresponding to the direct binding of $Au_{15}SG_{12}$-Ao to carbonylated amino acids on $W^{28}$, $W^{123}$ and $I^{98}$ —the same residues that we had identified as carbonylated in the previous analysis (Supplementary Fig. S3d). These results indicate that the $Au_{15}SG_{12}$-Ao is grafted directly on the carbonylated sites of the lysozyme. Of note, $S^{100}$, the carbonylated residue analyzed by molecular modeling (Fig. 2d) was not found among Ao-binding sites, likely due to degradation of the respective peptides.

**Application of $Au_{15}SG_{12}$-Ao for detection of carbonylated proteins in polyacrylamide gels.** We next tested whether labeling of carbonylated proteins with $Au_{15}SG_{12}$-Ao is applicable for the

**Table 1 List of oxidized peptides identified in MCO-lysozyme sample labeled with Ao-C3-Thiol-IAM after degradation of the Au15SG12-Ao NCs, using Protein Prospector.**

| m/z | z | Peptide + Ao-C3-Thiol-IAM modification | Error ppm | Score | Expect | Area in FMS |
|---|---|---|---|---|---|---|
| 399.8528 | 3 | CELAAAMKR[14][+148.0307] | −8.2 | 30.2 | 7.50E−04 | 4.15E+05 |
| 961.7669 | 3 | GYSLGNW[28][+148.0307]VCAAKFESNFNTQATNR | −2.6 | 41.2 | 7.50E−10 | 5.96E+05 |
| 961.7669 | 3 | GYSLGNWVCA[31][+148.0307]AKFESNFNTQATNR | −2.6 | 40.9 | 1.70E−09 | 5.44E+06 |
| 952.7581 | 3 | GYSLGNW[28][+121.0098]VCAAKFESNFNTQATNR | −8.1 | 46.2 | 2.50E−08 | 4.23E+05 |
| 952.0914 | 3 | GYSLGNW[28][+119.0041]VCAAKFESNFNTQATNR | −8.6 | 45 | 1.20E−07 | 2.21E+05 |
| 651.3103 | 3 | KI[98][+148.0307]VSDGNGMNAWVAWR | −8 | 25.9 | 1.30E−04 | 9.10E+05 |
| 388.1763 | 3 | GTDVQAW[123][+119.0041]IR | −14 | 30.2 | 7.50E−04 | 1.37E+04 |

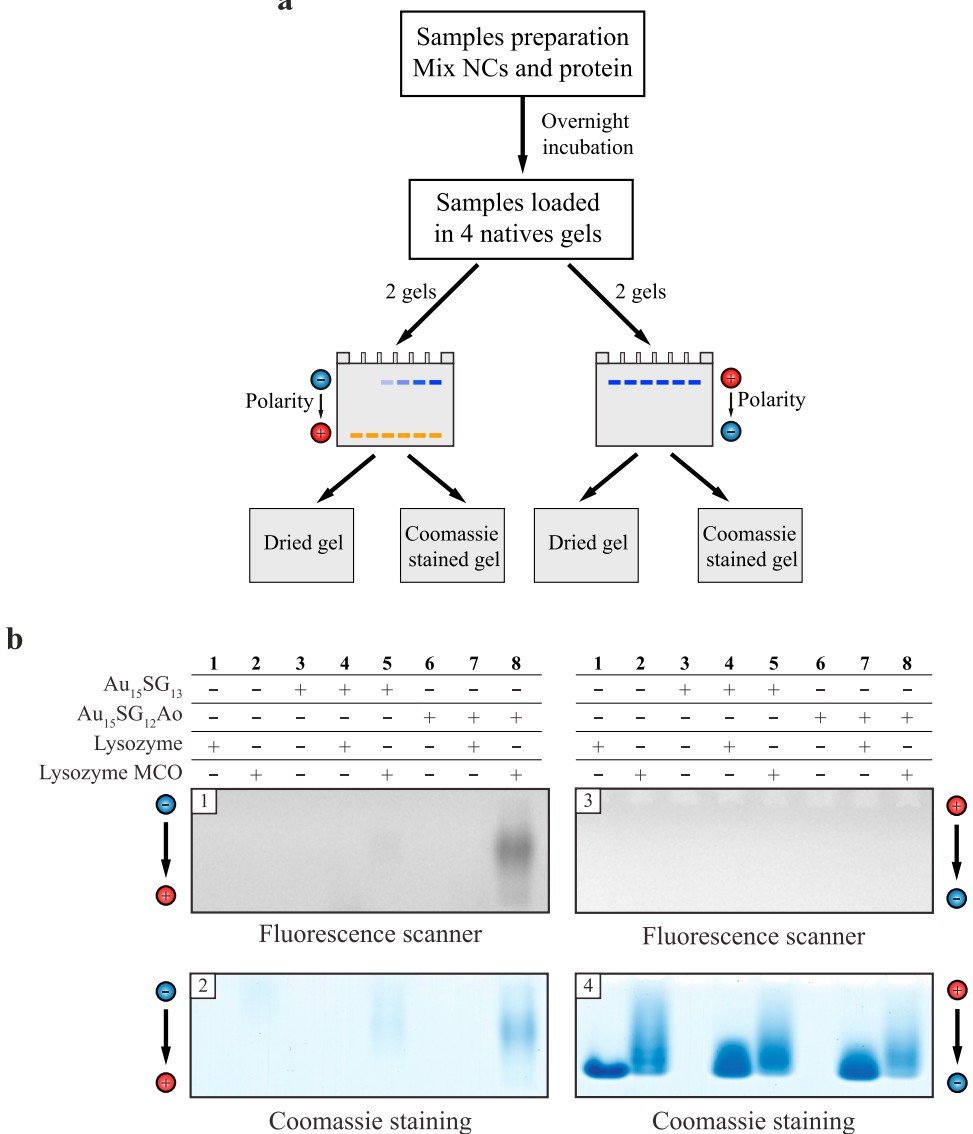

**Fig. 3 Au15SG12-Ao can detect carbonylated proteins in polyacrylamide gels. a** Schematic representation of the experimental setup for NCs migration in gels. **b** Four identical sets of samples were simultaneously migrated on native polyacrylamide gels with electricity direction from anode to cathode (gels 1 and 2) or from cathode to anode (gels 3 and 4). Gels 1 and 3 were dried and imaged by a fluorescence imaging scanner. Gels 2 and 4 were stained using Coomassie staining and images were obtained using a gel scanner. The displayed images are representative of three replicates. Source data and replicates for (**b**) are provided in Supplementary Data 1 Figs. S5-7.

detection of the protein–NC complexes in polyacrylamide gels. As the NCs are sensitive to detergents and can react with free thiols commonly present in the buffers used in sodium-dodecyl sulfate polyacrylamide gel electrophoresis (SDS-PAGE), native PAGE was applied in these experiments. As the first step, individual migration properties were determined for the NC and for

the oxidized lysozyme during native PAGE. To that end, we applied PAGE both with the conventional (migration direction from anode to cathode) and with the inverted polarity (migration direction from cathode to anode) (Fig. 3a, Supplementary Fig. S4 and Supplementary Data 1 Fig. S13). Visualization of the NC in the gels was achieved using fluorescence imaging, whereas the

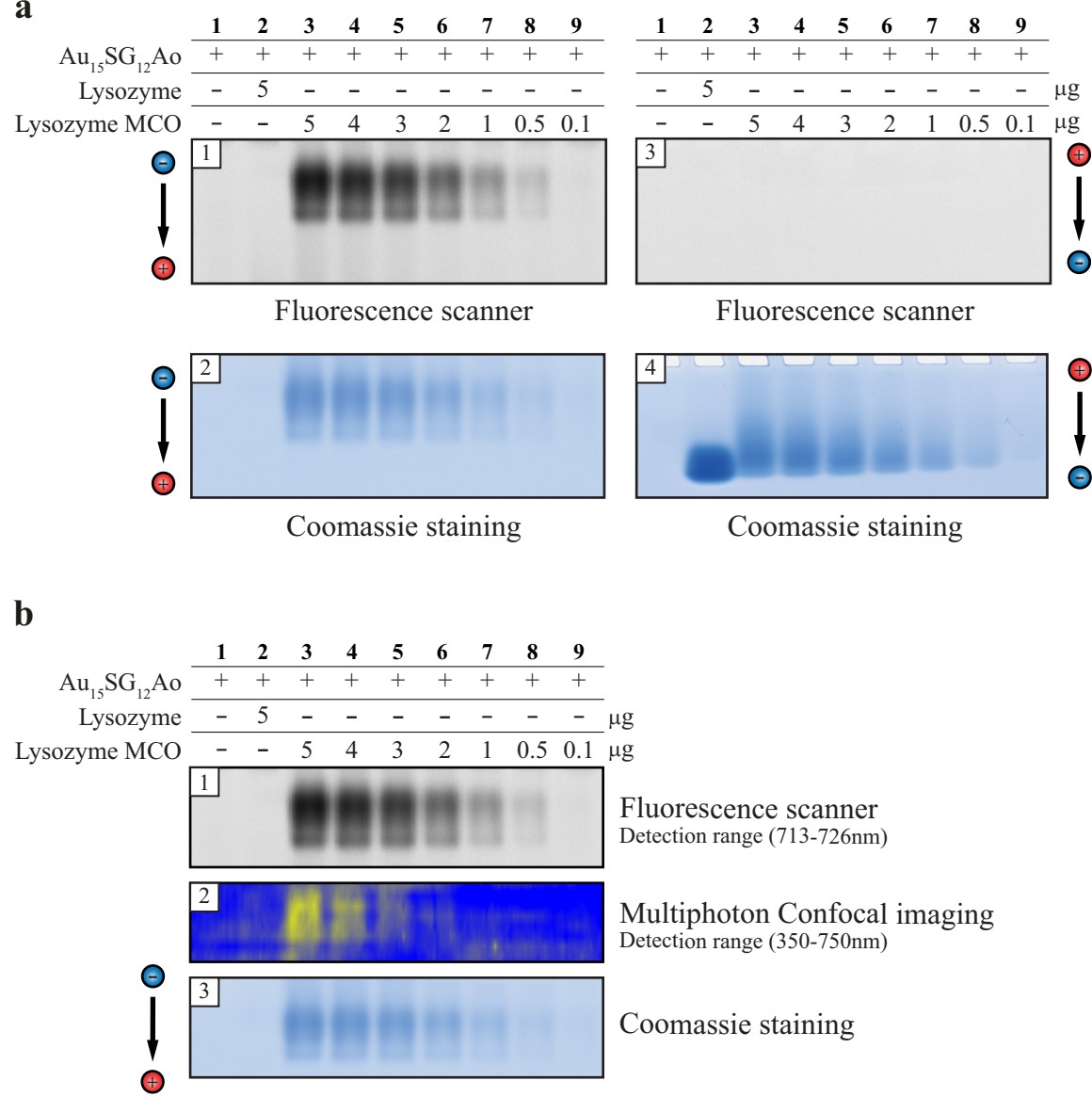

**Fig. 4 Au$_{15}$SG$_{12}$-Ao in-gel signal responds to the lysozyme quantity. a** The in-gel signal of Au$_{15}$SG$_{12}$-Ao decreases with the lowering of the oxidized lysozyme amounts. Gels 1 and 3 were dried and imaged using a fluorescence scanner. Gels 2 and 4 were stained using Coomassie staining and images were obtained using a gel scanner. The displayed images are representative of three replicates. **b** Comparison of Au$_{15}$SG$_{12}$-Ao for detection of carbonylated proteins in gels by one-photon excited fluorescence scanner and multiphotonic (two-photon excited fluorescence) confocal imaging. The displayed images are representative of three replicates. Source data and replicates for (**a**) are provided in Supplementary Data 1 Figs. S9–11 and source data for (**b**) is provided in Supplementary Data 1 Fig. S12.

proteins were visualized using Coomassie staining. Interestingly, NC and the lysozyme displayed opposing migration properties: while NC was detectable in the gel exposed to the conventional polarity, the lysozyme (both oxidized and non-oxidized) that has isoelectric point of 11.35 under given conditions migrated into the gel only upon inversion of the polarity (Supplementary Fig. S4). Hence, we subjected a mixture of Au$_{15}$SG$_{12}$-Ao with the oxidized lysozyme along with corresponding controls to native PAGE (Fig. 3b and Supplementary Data 1 Figs. S5-7), expecting to detect the stable complex formed between the protein and the NC in one of the two gels. Indeed, we observed co-migration of the Au$_{15}$SG$_{12}$-Ao and the oxidized protein, which was consistent with our result that NCs are grafted on the carbonylated sites of the protein (Fig. 3b, gels 1 and 2, lane 8). The complex was globally negatively charged since it migrated from anode to cathode (Fig. 3b, gels 1 and 2) and no fluorescence was observed

in the gel upon migration from cathode to anode (Fig. 3b, gels 3 and 4). Furthermore, there was no detectable complex of Au$_{15}$SG$_{12}$-Ao with the non-oxidized protein (Fig. 3b, gels 1 and 2, lane 7), indicating that there were no unspecific, carbonyl-unrelated interactions between the protein and the NC. Importantly, little to no fluorescence was observed upon native PAGE of the Au$_{15}$SG$_{13}$ - oxidized lysozyme mixture (Fig. 3b, lane 5), further supporting the absence of unspecific interactions between the NC and the protein (Table 1).

We next evaluated the potential of the NC for quantitative analysis of protein carbonyls. Fixed concentration of 500 μM Au$_{15}$SG$_{12}$-Ao was incubated with a decreasing range of concentrations of the lysozyme (50–1 μM corresponding to 5–0.1 μg protein loaded in the gel) and the reaction products were migrated on native PAGE (Fig. 4a and Supplementary Data 1 Figs. S9–11). An obvious decrease in the fluorescent signal

corresponding to NC correlated with decreasing amounts of oxidized lysozyme, indicating that NC-based assays can be developed for quantifying the amount of carbonyls on a protein. Finally, we tested the applicability of NCs in protein carbonylation detection in polyacrylamide gels using biphotonic confocal microscopy and observed an intense two-photon excited fluorescent signal for higher protein quantities (Fig. 4b and Supplementary Data 1 Fig. S12).

Together, these data establish a novel NC, $Au_{15}SG_{12}$-Ao, as suitable for specific detection of protein carbonyls by PAGE while offering photostability and biocompatible optical properties.

## Discussion

Herein, we conducted a proof of concept study for developing the first NC-based imaging system for protein carbonylation detection. To that end, we synthesized an NC with protein carbonyl-binding properties and demonstrated the ability of such NC to detect carbonylation of an oxidized model protein in gel-based analyses by one-photon fluorescence imaging and by two-photon excited fluorescence confocal imaging.

Furthermore, we have developed a protocol for an easy synthesis of atomically precise $Au_{15}$ NCs and for their functionalization with a thiolated Ao probe via simple ligand exchange. These NCs were highly reactive towards protein carbonyls and they formed a stable oxime bond between the aminooxy on the NC and the natural or oxidation-induced carbonyls on leupeptin and on lysozyme, respectively. Molecular modeling was conducted to evaluate the exposure of such oxime bond to solvents. This approach evidenced the protective effect on the oxime bond of glutathione ligands surrounding the thiolated Ao thus demonstrating the robustness of the linkage between the NC and carbonylated proteins. This protective effect of glutathione clearly adds value to the NC-based as compared to dye-based approaches for labeling carbonyls.

We also demonstrated that the functionalized NCs can act as one- and two-photon excitation fluorescence contrast agents for the detection of protein carbonyls in polyacrylamide gels. Importantly, we were able to detect NC-oxidized protein complexes by multiphoton microscopy, where both the excitation wavelength (~800 nm) and the detection of TPEF photons (up to 750 nm) can be in the NIR range. If applied in vivo, this strategy could offer definitive advantages over visible-range fluorescence, particularly the lack of interfering autofluorescence typical for biological molecules. Another advantage of using metal NCs as opposed to commonly used dyes is that they are biocompatible, soluble, photostable and likely to pass through the cellular membranes due to their small size. The noble AuNCs composed of a small number of atoms stabilized by a peptide ligand might thus be further developed as agents for the detection of carbonyls within cells and tissues.

Features of the NCs that give them tremendous potential for exploitation are their modular nature and versatility. These features provide room for improvement and opportunities for adaptation of the NCs for various purposes by optimization of their size and by the choice of ligands. In this proof-of-concept study, we have observed a correlation between the NC fluorescence and the amount of protein carbonyls, uncovering the potential of the NC for quantitative analyses. This NC could thus be further refined to gain higher sensitivity and to provide highly quantitative data. Moreover, to the best of our knowledge, this is the first study where the nonlinear optical properties of NC have been used to detect a post-translational modification on a protein. Given the extreme importance of post-translational modifications in life sciences, this pioneering study could lead to many alternative applications of the NCs.

In addition to the presented results, significant future applications of AuNCs may be in imaging of carbonylated proteins in fixed or live cells, allowing for quantification, as well as localization and transport studies of carbonylated proteins. Such studies advance understanding of the role of protein carbonylation in aging and ARDs.

## Methods

**$Au_{15}SG_{13}$ NCs synthesis**. Approximately, 235 mg of L-Glutathione (GSH) was dissolved under stirring conditions (at 45 °C) in 35 mL of methanol. 4 mL tributylamine was added to this mixture, which led to solubilization of GSH in methanol. This clear solution was further supplied with ~100 mg of $HAuCl_4.3H_2O$ previously dissolved in 4 mL water. The resulting solution was stirred for 5 min and further supplied with ~50 mg of trimethylamine borane ($TMA-BH_3$) after 2 h of stirring under 45–50 °C. The resulting solution was then stirred overnight at room temperature. After 24 h of stirring, the solution acquired a yellow color, and 1 mL $NH_4OH$ was added to induce precipitation. The resulting dispersion was centrifuged at 6000 rpm for 3–4 min. The supernatant was discarded and the pellet was redispersed in minimum water. The solution was supplied with methanol to induced further precipitation. The dispersion was centrifuged again at 6000 rpm for 3-4 min and the so obtained pellet was dissolved in 10 mL water and 2 mL of glacial acetic acid. The solution was left unperturbed for 4 h (minimum). This led to the precipitation of $Au_{10}$ NCs. The pellet was discarded and the supernatant was re-supplied with methanol to induce further precipitation. Following another cycle of precipitation, the pellet was dispersed in methanol and diethyl ether and dried overnight under vacuum.

**$Au_{15}SG_{12}$-Ao preparation**. $Au_{15}SG_{13}$ NCs were post functionalized via ligand-exchange reaction with aminooxypropanethiol as functional ligand using two methods. $Au_{15}SG_{13}$ was used as mother solution for both methods. In the method 1, a solution of Aminooxy corresponding to the desired quantities (0.1 to 1 equivalent relative to $Au_{15}SG_{13}$) was added to a water solution of $Au_{15}SG_{13}$ (1 mg/ml, pH ~ 8.5). The obtained solution was stirried at room temperature for 3 h (method 1). For method 2, the same protocol is used, except that aminooxy is added fractionally (0.1 equivalent of Aminooxy every 30 min) at ambient temperature or at 45 °C. It is worth mentioning that the products with varying numbers of aminooxypropanethiol could not be separated further. Instead, to purify the mixture of products from other NCs comprising of varying numbers of gold atoms, the product was precipitated with methanol/acetic acid solution.

**Oxidation protocols**. For metal-catalyzed oxidation (MCO), the protocol was based on Maisonneuve et al.[45]. Lysozyme from chicken egg white (Sigma) was dissolved in phosphate-buffered saline 1×, pH 7.4 (PBS–Roth) at 5 mg/ml. Oxidation was performed by supplementing 300 µL of protein solution (1.5 mg) with a freshly prepared mixture of ascorbic acid/$FeCl_3$ (Sigma/Kemika) with final concentrations of 25 mM/100 µM. 3 h incubation at RT was performed in a thermomixer at 500 rpm. Oxidation was stopped by the addition of 1 mM EDTA (Fluka) and cooling in ice.

**Carbonyl detection by Western Blotting**. For the Western Blot, carbonyls were derivatized with 10 mM EZ-Link™ Alkoxyamine-PEG4-Biotin (Thermofisher) for 3H at RT. Samples were subjected to gel electrophoresis using the Mini-Protean® Tetra Cell system (Bio-Rad). Gels were cast homemade using Acrylamide/bisacrylamide (Fisher Bioreagent), APS (Biosolve), TEMED (Sigma) and Tris-glycine buffer. Linear gels (20%) were cast and used for sample migration. Proteins were then transferred to a PVDF membrane the Trans-Blot Turbo Transfer System (BioRad) with 25 V constant (up to 1.0 A) for 30 min. Membranes were stained with Red Ponceau dye (Sigma) to assess transfer efficiency and total protein loading. Membranes were then blocked with TBS-Tween 0.05%–Milk 5% buffer for 30 min at room temperature with shaking. Next, membranes were incubated for 1h at RT with Streptavidin-Alexa Fluor 700 (Invitrogen) probes resuspended. At every step after blocking, membranes were washed 4 times 5 min with TBS-Tween 0.05%. Finally, Typhoon™ FLA 9500 biomolecular imager (GE Healthcare) was used to measure fluorescence. Experiments were performed in triplicates. All quantifications were performed using ImageLab software (Bio-rad) and the statistical analysis was performed in GraphPad Software.

**MS proteomics. Sample preparation**. Prior to LC-MS/MS analysis, the MCO protein was grafted with DNPH. Proteins samples were derivatized with 10 mM of DNPH (final concentration) at RT for 30 min with shaking (500 rpm). Neutralization of the reaction was done using 1 M Ammonium bicarbonate solution to reach pH 8.

Also, MCO protein grafted with the aminooxy NC was degraded with cysteine. 300 µL of 500 µg of MCO lysozyme labeled with an excess of $Au_{15}SG_{12}$-Ao were diluted in 700 µL of a 10 mM NaCl aqueous solution. Then, 200 µL of acetone was added to neutralize ungrafted aminooxy groups. Solution was left overnight before starting the NC degradation. For this, we used a large excess of cysteine by adding

100 μL of cysteine (10 mM). The solution was sonicated 45 min and 1 mL of methanol was added before another 45 min of sonication. Then, 50 μL of glacial acetic acid was added to complete precipitation of gold-cysteine polymer. The precipitate was removed by centrifugation (11000 rpm/10 min) and the supernatant was evaporated under vacuum. The lysozyme was redispersed in 0.5 mL of water before sample preparation for MS analysis (i.e. reduction, alkylation, and digestion).

Derivatized protein samples were then reduced in 8 M urea, 15 mM dithiothreitol (DTT) at 60 °C for 40 min, and then alkylated with 35 mM iodoacetamide (IAM) at room temperature in the dark for 40 min. To reduce the urea concentration, the samples were diluted 5-fold with ammonium bicarbonate (AMBIC) before overnight digestion at 37 °C with trypsin (type IX-S from Porcine Pancreas) using a 1:30 (w/w) enzyme to substrate ratio. Digestion was stopped by the addition of formic acid (FA) to a final concentration of 0.5%.

All samples were desalted and concentrated using Oasis HLB 3cc (60 mg) reversed-phase cartridges (Waters, Milford, MA, USA) (elution with 1.5 mL of methanol containing 0.5% FA). All samples were evaporated to dryness and resuspended in 150 μL of water/acetonitrile (ACN) (90:10, v/v) containing 0.5% FA. All solutions were stored at −18 °C before use.

**MS Proteomics. Instrumentation and Operating Conditions**. Mass spectrometry analyses were performed on a hybrid quadrupole-orbitrap Q-Exactive® mass spectrometer (Thermo Fisher Scientific, San Jose, CA, USA) equipped with a HESI ion source coupled to a Surveyor HPLC-MS pump (Thermo Fisher Scientific, San Jose, CA, USA) and a PAL Auto-sampler (CTC Analytics, Switzerland).

The HPLC separation was carried out on an XBridge C18 column (100 × 2.1 mm, 3.5 μm) from Waters. The HPLC mobile phase consisted of water containing formic acid 0.1% (v/v) as eluent A, and ACN containing formic acid 0.1% (v/v) as eluent B. Elution was performed at a flow rate of 300 μL/min. The elution sequence, for the digested protein samples, included a linear gradient from 10% to 60% of eluent B for 52 min, then a plateau at 95 % of eluent B for 4 min. The gradient was returned to the initial conditions and held there for 4 min. The injection volume was 10 μL.

Ionization was achieved using electrospray in the positive ionization mode with an ion spray voltage of 4 kV. The sheath gas and the auxiliary gas (nitrogen) flow rates were respectively set at 35 and 10 (arbitrary unit) with a HESI vaporizer temperature of 400 °C. The ion transfer capillary temperature was 300 °C with a sweep gas (nitrogen) flow rate at 5 (arbitrary unit). The S-lens RF was set at 90 (arbitrary unit). The Automatic Gain Control (AGC) target was $3 \times 10^6$ and the maximum injection time was set at 250 ms. Experiments were done in data-dependent top 10 modes. The full MS scans were done over an $m/z$ 300-1500 range with a resolution of 35000. For the data-dependent MS/MS scans, the resolution was set at 17500, isolation 2 $m/z$, with a normalized collision energy of 28 (arbitrary unit). To exclude the redundant processing of dominant ions and allow selection of low abundant oxidized peptides, a dynamic exclusion time of 20 s was set.

**MS proteomics. Peptide and protein identification and quantification**. Fragmentation data were converted to peak lists using PAVA RawRead and searched against sequences of Gallus Gallus (Chicken) proteins contained in the Swissprot human database (downloaded 2017.11.01, 556006 entries) using Protein Prospector[46]. All searches used the following parameters: mass tolerances in MS and MS/MS modes were 20 ppm and 0.2 Daltons, respectively. Trypsin was designated as the enzyme and up to two missed cleavages were allowed. Carbamidomethylation of cysteine residues was designated as a fixed modification. The considered standard variable modifications were N-terminal acetylation, N-terminal glutamine conversion to pyroglutamate and methionine oxidation. The maximum allowed expected value was set at up to 0.01 (protein) and 0.05 (peptide). To search for oxidized peptides, user-defined variable modifications, corresponding to a list of 43 known carbonyl modifications from the literature[9,42] derivatized with DNP (Oxi-DNP) or after degradation of the $Au_{15}SG_{12}$-Ao NC (Ao-C3-thiol-IAM), were implemented. All peptides identified in a top10 analysis have been quantified by using the MS1 filtering tool in Skyline. The peptides have been integrated allowing a match tolerance of 0.055 $m/z$ and a minimum isotope dot product of 0.9. All integrations have been verified manually and the total area of each peptide has been reported for the most intense charge states.

**Experimental setup for protein carbonyls detection with NCs in 1D gel electrophoresis**. After oxidation of the recombinant protein with MCO protocol as described previously, recombinant protein and NCs were resuspended in PBS 1X and were incubated at a final concentration of 50 μM (Fig. 3 and Fig. 4) and 500 μM, respectively, at 10°C overnight in a rotating shaker. The concentration of protein and NCs were modified to 137 μM and 50 μM respectively for multiphoton confocal imaging. Samples were then supplemented with 10% glycerol, loaded in 15% or 20% homemade Tris-Glycine gels, and migrated with Tris-glycine buffer 1× (25–192 mM). Migration polarity was done as indicated in the figures. Before drying, gels were equilibrated in a 20% ethanol/5% glycerol solution for 20 min and then placed in a drying frame (Serva) for a minimum of 48 h. Typhoon™ FLA 9500 biomolecular imager (GE Healthcare) was used to detect fluorescence of these gels as described below. Protein staining in the gel was performed using a Coomassie staining solution (Ammonium sulfate 10%–phosphoric acid 10%–Coomassie G250 0.12% and ethanol 20%) overnight with shaking at room temperature followed by destaining with distilled $H_2O$. Gels were scanned with the BIO-5000 Plus VIS Gel Scanner from Serva. All experiments were performed in triplicate.

**One-photon fluorescence setup**. One-photon fluorescence measurements were performed with Typhoon™ FLA 9500 biomolecular imager using a 473 nm (blue LD laser—for NCs detection) or a 685 nm laser (Red LD laser—for AlexaFluor700 detection) for excitation and a BPFR700 (R715) filter to collect the emitted fluorescence in the wavelength range from 713 nm to 726 nm.

**Two-photon fluorescence setup**. Two-photon fluorescence measurements were performed with a customized confocal microscope (TE2000-U, Nikon Inc.) in which the excitation light entrance has been modified to allow free-space laser beam input, instead of the original optical-fiber light input. The luminescence was excited at 780 nm with a mode-locked frequency-doubled femtosecond Er-doped fiber laser (C-Fiber 780, MenloSystems GmbH). The laser spectrum was bounded by two filters (FELH0750 and FESH0800, Thorlabs Inc.). The output power of the femtosecond laser was 62 mW. The laser beam was focused by a Nikon Plan Fluor Ph1 DLL objective (10×/0.30 NA). The sample was XY scanned by the inner microscope motorized stage and galvanoscanner (confocal C1 head, Nikon Inc.), and the Z scan was performed by the inner microscope motorized focus. The emitted signal was collected in epifluorescence illumination mode. The two-photon fluorescence emission was separated from the incident light through a dichroic mirror (NFD01-785, IDEX Health & Science LLC). A FESH0750 filter was used in order to remove the photons coming from the excitation laser and collect visible 350-750 nm fluorescence on the inner microscope photomultiplier tube. TPEF intensity raster scans performed at several Z positions of the gel (size of the gel image: 60 × 48 mm). Time per each point (0.25 mm×1 mm): (61 μs × 2 × 128 × 128 for averaging).

**Computational**. In order to determine the structural properties of $Au_{15}SG_{12}$-Ao liganded cluster and protein-liganded cluster the following procedures have been used. First, QM/MM method within ONIOM two layer[47–49] approach implemented in Gaussian[50] has been employed for $Au_{15}SG_{12}$-Ao liganded cluster. $Au_{15}$, sulfur atoms and 3-(Aminooxy)-1-propanethiol have been included in QM. For the gold atoms the 19-e⁻ relativistic effective core potential (19-e⁻ RECP) from the Stuttgart group[51] taking into account scalar relativistic effects has been used. For atoms within QM, split valence polarization atomic basis sets (SVP)[52] and the hybrid B3LYP functional[53–56] have been employed. In MM part UFF force field[57] has been employed for all ligands, with exception of 3-(Aminooxy)-1-propanethiol. In order to analyze the hydrogen bonding network the obtained ONIOM two layer B3LYP/UFF structures have been reoptimized using SEQM PM7[58] where gold and sulfur atoms have been frozen. Notice that Au-Au distances are overestimated within PM7 approach[59]. Two isomers of $Au_{15}SG_{12}$-Ao have been obtained using PM7 and single point DFT (B3LYP) calculations as shown in Supplementary Fig S2g. The structure in Fig. 2d including Lysozyme has been obtained by QM/MM approach. QM part is treated by PM7 and describes interface between liganded cluster and protein. It contains $Au_{15}$, sulfur atoms from glutathiones and Ao-Serine bond. The other ligands and the rest of the protein have been included in MM where UFF force field was employed. In order to simulate penetration of water molecules into $Au_{15}SG_{12}$-Ao-Lysozyme 800 neutral $H_2O$ molecules have been added to MM part within QM/MM approach for the optimization of geometry. In addition, distances from Ao-Serine100 bond to each water molecule was taken into account in order to obtain the radial distribution of the density of water molecules using R-studio software[60] (cf Supplementary Fig. S3f).

## Data availability

Data available on request from the authors.

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

## Acknowledgements

This research was supported by the project STIM-REI, Contract Number: KK.01.1.1.01.0003, funded by the European Union through the European Regional Development Fund—the Operational Programme Competitiveness and Cohesion 2014–2020 (KK.01.1.1.01). V.B.K., M.P.B., and H.F. acknowledge computational facilities of the HPC computer within the STIM-REI project, Doctoral study of Biophysics at University of Split as well as Prof. Miroslav Radman at MedILS and Split-Dalmatia County for support. M.G. acknowledge the French National Research Agency for the financial support (Grant Agreement ANR-18-CE29-0002-01 HyLOxi). We would also

like to acknowledge the financial support received from the French-Croatian project "International Laboratory for Nano Clusters and Biological Aging, LIA NCBA". The authors would like to thank Céline Brunon and Esther Jarrossay from Science et Surface (www.science-et-surface.fr) for XPS and FTIR spectra.

## Author contributions

R.A., V.B.-K., M.R., and A.K. conceived the initial idea and coordinated the work. F.B. synthesized and prepared the NCs, assisted with S.B. H.F. recorded and analyzed mass spectra. C.M. conducted nonlinear optics imaging, supervised by P.-F.B. and I.R.-A. M.P.B., Z.S.M., and V.B.-K. performed and analyzed the theoretical results. M.G. conducted proteomics-based MS protocols, measurements and analysis. G.F.C. developed gel strategies, assisted with R.L. and performed fluorescence measurements supervised by K.T. R.A., P.D., M.R., and V.B.-K. supervised and financed the project. R.A., G.F.C., K.T., and V.B.-K. wrote the paper. All authors provided critical feedback and helped to shape the final manuscript.

## Competing interests

The authors declare no competing interests.
