## [Peer Review File · Communications Chemistry]

Reviewers' comments:

Reviewer #1 (Remarks to the Author):

The authors have developed a novel strategy to design AuNCs for precise detection a causal ageing biomarker of protein carbonylation. The authors have synthesized Au15SG13 and further functionalized it with a thiolated aminoxy moiety to gain protein carbonyl-binding properties. The authors have also demonstrate that Au15SG12-Aminoxy detects protein carbonylation in gel-based 1D electrophoresis by one- and two-photon excited fluorescence. Overall, this work can inspire more AuNC design ideas for the application in nonlinear optical probe. Therefore, I would like to recommend this work to publish in Communications Chemistry. The following issues should be addressed to further improve the quality of this work.

1. To demonstrate the successful synthesis of Au15SG13, TEM image of Au15SG13 should be provided.
2. FTIR spectra of SG and Au15SG13 should be provided to prove the conjugation between gold and SG.
3. XPS spectra of Au15SG13 should be provided to demonstrate the successful synthesis of Au15SG13
4. More references should be cited to broaden the introduction more completely in the utilization of AuNCs for in vitro cell labelling and in vivo fluorescence imaging applications. (Anal. Chem. 2018, 90, 3974–3980; ACS Sustainable Chem. Eng. 2019, 7, 15479–15486)

Reviewer #2 (Remarks to the Author):

Combes et al. have used Au15SG13 NCs for precise detection of protein carbonylation. First, the Au15SG13 NCs were ligand exchanged with a thiolated aminoxy, and then that was used as a biomarker probe. The materials were characterized precisely using HRESI MS. The manuscript is interesting, and I can recommend it for publication after major revision. Bellows are some comments and suggestions which need to be taken care of in the revised version.

1. The authors used different nomenclature throughout the paper (such as NCs, Nanocluster, cluster, etc.). I think it would be good to use nanoclusters (NCs) for the first time it appeared in the MS, and after that, they can use only 'NCs.'
2. Figure 1: The ESI MS of Au15SG13 and Au15SG12Ao (b and d) can be plotted in the same graph. The first full-range ESI MS spectrum can be moved to SI. The HRESI MS isotopic pattern can be moved from SI to this figure 1.
3. The author can consider adding an elaborate discussion about the concentration-dependency of the ligand exchange of Au15SG13 with Ao (details should be added in the synthesis). The authors can add a concentration-dependent ESI MS experiment and explain the controlling of ligand-exchange to just one at some specific concentration of Ao. As ligand-exchange is always a dynamic process, it is a very crucial step for the applications.
4. Since the author already has the DFT structures, they can be used the computed structures in Figure 1 instead of the scheme. Can the author explain why one ligand-exchange is more feasible here? If H-bonding is observed in the most stable isomer, then more and more H-bonds (more than one ligand-exchanged case) can help in extra stabilization?
5. The author explains the H-bonding of Au15SG12Ao with proteins also. Are there any competitive binding preferences (H-bond between NCs over protein)? It looks like glutathione are majorly responsible for the H-bonds (out of the 9 H bonds), then most of the protein binding job can be done by glutathione on NCs? Meaning Au15SG13 will be better effective than Au15SG12Ao?

Reviewer #3 (Remarks to the Author):

In this paper, Combes et al. use the Au₁₅(SG)₁₂-Aminoxy gold nanoclusters to detect protein carbonylation. The detection of post-translational protein modification with gold-thiolate clusters is (to my knowledge) novel and demonstrates a potentially very useful and interesting use of these gold materials. The experimental procedures are outside my expertise and I will therefore not comment on those. The computational methods used in this paper are overall appropriate although a few things should be clarified:

1) The optimization of the NC was done using a two-layer QM/MM approach. The QM region consists of Au atoms, S atoms and the full Au ligand. Were the other ligands modeled with PM7 or MM? If MM, what force field? It is unclear from the computational details section.

2) The clusters were reoptimized with PM7 to analyze the hydrogen bonding network. Was the ΔE for the two isomers computed with PM7 or DFT? I believe DFT single-point energy calculations can be performed on these clusters to get their relative energies.

3) Also, the authors need to keep in mind that PM7 overestimates Au-Au distances [J. Phys. Chem. A 2020, 124, 13, 2601–2615] Were the gold atoms frozen at their DFT coordinates?

Dear Reviewers:

We thank you for your valuable comments concerning our manuscript entitled “Golden Touch: Functionalized Au₁₅ Nanoclusters as Luminescent Probes for Protein Carbonylation Detection” (ID: COMMSCHEM-20-0425-T). Your comments are all valuable and very helpful for revising and improving our paper, as well as the important guiding significance to the readers for this research. Revised portions are marked in yellow in the paper. The main corrections in the paper and the response to the reviewer’s comments are as following:

Reviewer #1 (Remarks to the Author):

The authors have developed a novel strategy to design AuNCs for precise detection a causal ageing biomarker of protein carbonylation. The authors have synthesized Au₁₅SG₁₃ and further functionalized it with a thiolated aminoxy moiety to gain protein carbonyl-binding properties. The authors have also demonstrate that Au₁₅SG₁₂-Aminoxy detects protein carbonylation in gel-based 1D electrophoresis by one- and two-photon excited fluorescence. Overall, this work can inspire more AuNC design ideas for the application in nonlinear optical probe. Therefore, I would like to recommend this work to publish in Communications Chemistry. The following issues should be addressed to further improve the quality of this work.

1. To demonstrate the successful synthesis of Au₁₅SG₁₃, TEM image of Au₁₅SG₁₃ should be provided.
2. FTIR spectra of SG and Au₁₅SG₁₃ should be provided to prove the conjugation between gold and SG.
3. XPS spectra of Au₁₅SG₁₃ should be provided to demonstrate the successful synthesis of Au₁₅SG₁₃

Response: First of all, we would like to point out that the materials (Au₁₅SG₁₃, and Au₁₅SG₁₂AO) were characterized precisely using ESI-HRMS. However, we agree with the reviewer that additional characterization (in particular permitting to get more chemical information) on “precursor” Au₁₅SG₁₃ nanoclusters were required. We thus performed additional XPS, FTIR and TEM analysis (presented in Supplementary Information) and added the following paragraph in the revised version:

FT-IR spectra of Au₁₅SG₁₃ NCs and pure glutathione GSH are given in supplementary Figs. S1a and S1b). The ligation of glutathione in the form of the thiolate (SG) to the Au core was confirmed by the absence of the absorption band at $\nu(\text{S-H}) = 2523 \text{ cm}^{-1}$ in the FTIR spectrum of the as-prepared NCs sample, as already found in the seminal work published by Negishi and Tsukuda.³⁵ Supplementary Fig. S1c shows the TEM image of the as-prepared Au₁₅SG₁₃ NCs. The particles with the sizes of 1-2 nm are barely discernible in the image. From the XPS data (see in supplementary Table S1), we find the Au/S atomic ratio to be 1.26 ± 0.13 , which is compatible with the composition of Au₁₅SG₁₃ (the expected value is 1.15).

4. More references should be cited to broaden the introduction more completely in the utilization of AuNCs for in vitro cell labelling and in vivo fluorescence imaging applications.
(Anal. Chem. 2018, 90, 3974–3980; ACS Sustainable Chem. Eng. 2019, 7, 15479–15486)

Two recent reviews :

1. Porret E, Le Guevel X, Coll JL. Gold nanoclusters for biomedical applications: toward in vivo studies. *J Mat Chem B* 8, 2216-2232 (2020).
2. Bai YL, Shu T, Su L, Zhang XJ. Fluorescent Gold Nanoclusters for Biosensor and Bioimaging Application. *Crystals* 10, 12 (2020).

And the two suggested papers:

1. Chang T-K, et al. Metabolic Mechanism Investigation of Antibacterial Active Cysteine-Conjugated Gold Nanoclusters in *Escherichia coli*. *ACS Sustainable Chemistry & Engineering* 7, 15479-15486 (2019).
2. Cheng T-M, et al. Quantitative Analysis of Glucose Metabolic Cleavage in Glucose Transporters Overexpressed Cancer Cells by Target-Specific Fluorescent Gold Nanoclusters. *Analytical Chemistry* 90, 3974-3980 (2018).

Have been added after the sentence:

"In addition, ligated AuNCs exhibit outstanding biocompatibility, therefore, their utilization for in vitro cell labelling and in vivo fluorescence imaging applications has been a rich research area."

Reviewer #2 (Remarks to the Author):

Combes et al. have used Au15SG13 NCs for precise detection of protein carbonylation. First, the Au15SG13 NCs were ligand exchanged with a thiolated aminoxy, and then that was used as a biomarker probe. The materials were characterized precisely using HRESI MS. The manuscript is interesting, and I can recommend it for publication after major revision. Below are some comments and suggestions which need to be taken care of in the revised version.

1. The authors used different nomenclature throughout the paper (such as NCs, Nanocluster, cluster, etc.). I think it would be good to use nanoclusters (NCs) for the first time it appeared in the MS, and after that, they can use only 'NCs.'

Response: We thank the reviewer for the useful comment. In the revised version, we now use the same nomenclature throughout the paper (nanoclusters (NCs)).

2. Figure 1: The ESI MS of Au15SG13 and Au15SG12Ao (b and d) can be plotted in the same graph. The first full-range ESI MS spectrum can be moved to SI. The HRESI MS isotopic pattern can be moved from SI to this figure 1.

Response: Good suggestion. Fig. 1 has been changed (b and d became c) and the first full-range ESI MS was moved from Figure 1 to SI (Fig. S2a). HRESI MS isotopic pattern was kept in SI (Fig. S2b,c).

3. The author can consider adding an elaborate discussion about the concentration-dependency of the ligand exchange of Au15SG13 with Ao (details should be added in the synthesis). The authors can add a concentration-dependent ESI MS experiment and explain the controlling of ligand-exchange to just one at some specific concentration of Ao. As ligand-exchange is always a dynamic process, it is a very crucial step for the applications.

Response: We thank the reviewer for the this very constructive comment. ESI-MS was applied to monitor the number of Ao ligand exchanged in Au₁₅SG₁₃ species following addition of Ao in solution at different concentrations. The following figure shows the evolution in ligand exchange as a function of the concentration of Ao ligand (in terms of Ao eq value) using different methods. Clearly, adding 0.1 – 0.3 eq of Ao allows for controlling of ligand-exchange to just one.

These figures (Fig. S2d,e) and a small paragraph (describing the methods) were added in the revised version of the manuscript and its supplementary information.

4. Since the author already has the DFT structures, they can be used the computed structures in Figure 1 instead of the scheme.

Response: Good suggestion. Done.

Can the author explain why one ligand-exchange is more feasible here? If H-bonding is observed in the most stable isomer, then more and more H-bonds (more than one ligand-exchanged case) can help in extra stabilization?

Response: The reviewer is right. More than one ligand-exchange can help in extra stabilization and in more efficient targeting to proteins. However, from the perspective of quantification of oxidized protein, we need to keep the ratio between nanoclusters and oxidized proteins at 1:1.

5. The author explains the H-bonding of Au₁₅SG₁₂Ao with proteins also. Are there any competitive binding preferences (H-bond between NCs over protein)? It looks like glutathione are majorly

responsible for the H-bonds (out of the 9 H bonds), then most of the protein binding job can be done by glutathione on NCs? Meaning Au₁₅SG₁₃ will be better effective than Au₁₅SG₁₂AO?

Response: According to the DFT calculations, binding energy for Aminoxy-propanethiol bound to the seryl through covalent (oxime) bond is 18 eV. In contrast, binding energy of H-bonds for Glutathione and Serine is 0.8 eV involving two H-bonds. Even larger number of H-bonds would not reach the energy of the covalent (oxime) bond. This means that Au₁₅SG₁₃ should not be more effective.

Reviewer #3 (Remarks to the Author):

In this paper, Combes et al. use the Au₁₅(SG)₁₂-Aminoxy gold nanoclusters to detect protein carbonylation. The detection of post-translational protein modification with gold-thiolate clusters is (to my knowledge) novel and demonstrates a potentially very useful and interesting use of these gold materials. The experimental procedures are outside my expertise and I will therefore not comment on those. The computational methods used in this paper are overall appropriate although a few things should be clarified:

Response: Thank you for pointing out unclear parts. Corrections have been introduced as detailed below.

1) The optimization of the NC was done using a two-layer QM/MM approach. The QM region consists of Au atoms, S atoms and the full AO ligand. Were the other ligands modeled with PM7 or MM? If MM, what force field? It is unclear from the computational details section.

Response: In the enclosed Computational part we introduced the following : “In MM part UFF force field⁵⁶ has been employed for all ligands, with exception of 3-(Aminoxy)-1-propanethiol.”

2) The clusters were reoptimized with PM7 to analyze the hydrogen bonding network. Was the ΔE for the two isomers computed with PM7 or DFT? I believe DFT single-point energy calculations can be performed on these clusters to get their relative energies.

Response: We also performed single point DFT (B3LYP/def2svp) calculations for two isomers of Au₁₅(SG)₁₂(3-(Aminoxy)-1-propanethiol) (compare Fig. S2g) and ΔE is 0.86 eV as shown in Fig. S2g.

3) Also, the authors need to keep in mind that PM7 overestimates Au-Au distances [J. Phys. Chem. A 2020, 124, 13, 2601–2615] Were the gold atoms frozen at their DFT coordinates?

Response: Yes, we are aware that PM7 overestimates Au-Au distances and we included this statement in computational part with the proposed reference:

“Notice that Au-Au distances are overestimated within PM7 approach⁵⁸.”

Gold atoms were frozen at their DFT coordinates and this has been added in computational part:

“In order to analyze hydrogen bonding network the obtained ONIOM two layer B3LYP/UFF structures have been reoptimized using SEQM PM7⁵⁷ where gold and sulfur atoms have been frozen.”

Enclosed is modified Computational part:

“ **Computational**

In order to determine the structural properties of Au₁₅SG₁₂-Ao ligated cluster and protein-ligated cluster the following procedures have been used. First, QM/MM method within ONIOM two layer⁴⁷⁻⁴⁹ approach implemented in Gaussian⁵⁰ has been employed for Au₁₅SG₁₂-Ao ligated cluster. Au₁₅, sulfur atoms and 3-(Aminoxy)-1-propanethiol have been included in QM. For the gold atoms the 19-e⁻ relativistic effective core potential (19-e⁻ RECP) from the Stuttgart group⁵¹ taking into account scalar relativistic effects has been used. For atoms within QM, split valence polarization atomic basis sets (SVP)⁵² and the hybrid B3LYP functional⁵³⁻⁵⁶ have been employed. In MM part UFF force field⁵⁷ has been employed for all ligands, with exception of 3-(Aminoxy)-1-propanethiol. In order to analyze hydrogen bonding network the obtained ONIOM two layer B3LYP/UFF structures have been reoptimized using SEQM PM7⁵⁸ where gold and sulfur atoms have been frozen. Notice that Au-Au distances are overestimated within PM7 approach⁵⁹. Two isomers of Au₁₅SG₁₂-(3-(Aminoxy)-1-propanethiol) have been obtained using PM7 and single point DFT (B3LYP) calculations as shown in Supplementary Fig S2g. The structure in Fig. 2d including Lysozyme has been obtained by QM/MM approach. QM part is treated by PM7 and describes interface between ligated cluster and protein. It contains Au₁₅, sulfur atoms from glutathiones and Ao-Serine bond. The other ligands and the rest of the protein have been included in MM where UFF force field was employed. In order to simulate penetration of water molecules into Au₁₅SG₁₂-Ao-Lysozyme 800 neutral H₂O molecules have been added to MM part within QM/MM approach for the optimization of geometry. In addition, distances from Ao-Serine100 bond to each water molecule were taken into account in order to obtain radial distribution of the density of water molecules using R-studio software⁶⁰ (cf Supplementary Fig. S3f).

”

REVIEWERS' COMMENTS:

Reviewer #1 (Remarks to the Author):

The authors have addressed all issues raised by the reviewers. Therefore, I would like to recommend this manuscript published as its current form in Communications Chemistry.

Reviewer #2 (Remarks to the Author):

All comments were taken care of in the revised version. The manuscript can be accepted in its present form.

Reviewer #3 (Remarks to the Author):

Reviewers suggestions were properly addressed. I recommend this paper for publication.